# Mechanisms of electrochemical hydrogenation of aromatic compound mixtures over a bimetallic PtRu catalyst
Cesar Catizane [1], Ying Jiang [1,2] ✉ & Joy Sumner [1] ✉

Efficient electrochemical hydrogenation (ECH) of organic compounds is essential for sustainability, promoting chemical feedstock circularity and synthetic fuel production. This study investigates the ECH of benzoic acid, phenol, guaiacol, and their mixtures, key components in upgradeable oils, using a carbon-supported PtRu catalyst under varying initial concentrations, temperatures, and current densities. Phenol achieved the highest conversion (83.17%) with a 60% Faradaic efficiency (FE). In mixtures, benzoic acid + phenol yielded the best performance (64.19% conversion, 74% FE), indicating a synergistic effect. Notably, BA consistently exhibited 100% selectivity for cyclohexane carboxylic acid (CCA) across all conditions. Density functional theory (DFT) calculations revealed that parallel adsorption of BA on the cathode (−1.12 eV) is more stable than perpendicular positioning (-0.58 eV), explaining the high selectivity for CCA. These findings provide a foundation for future developments in ECH of real pyrolysis oil.

Pyrolysis is the thermal decomposition of materials at high temperatures in an inert atmosphere. When applied to waste and biomass feedstocks, the condensable liquid product from pyrolysis (constitutes around 70 wt.% of the pyrolysis products), known as bio-oil is widely recognised as a low-carbon biofuel and precursor for chemical production. In the waste industry, there is significant interest in applying pyrolysis as an enabling technology for plastic polymer recycling processes, producing pristine polymers as platform chemicals. This can potentially address the global issue of low levels of plastic recycling; currently standing at less than 10% in the European Union, most of which uses mechanical recycling routes (i.e. limited to thermoplastics)[1–3].

Despite these potentials, bio-oil, is a complex mixture containing over 300 compounds which poses significant challenges for its practical use[4]. Its physical and chemical properties, including viscosity, C:H C:O ratios, and thermal and chemical stability, typically cannot meet the standards of direct application as a biofuel or as feedstock for the chemical and petrochemical industries[5]. Thus, upgrading bio-oil is essential to enhance its suitability for widespread use.

A key step to improving bio-oil quality for fuel and polymer recycling applications is to increase the hydrogen-to-carbon ratio crucial for stability through hydrogenation processes. Conventional hydrogenation technologies, including catalytic cracking and hydroprocessing. Both processes involve high temperature and pressure reactions. To date, commercial applications of these hydrogenation methods are rare due to the significant

operational costs, catalyst deactivation and undesirable secondary reactions, particularly tar formation[5–9]. Electrochemical hydrogenation (ECH), by contrast, can be performed under milder reaction conditions at low temperatures (below 80 °C) and ambient pressure to reduce reactive functionalities. Additionally, ECH process requires no $H_2$ supply to the process, therefore eliminating the costs and risks associated with hydrogen storage. These advantages make ECH a promising alternative to conventional routes for sustainable bio-oil upgrading[10–12].

However, as a proof-of-concept currently, fundamental reaction kinetics and mechanisms of key bio-oil constituent compounds during ECH process are still not fully understood. As such, this study uses key compounds, including guaiacol, phenol, benzoic acid and their derivatives commonly found in both biomass and plastic-derived bio-oil[13–15] as a model compound, to investigate their reaction mechanisms in an H-type cell set-up.

H-type cell is the most common design for electrochemical reactions. It consists of an anode and a cathode chamber, separated by a porous membrane (Fig. 1). Cathode material selection is currently one of the most important steps in ECH research and deployment, as it can influence the conversion rate, Faradaic efficiency (FE), and selectivity. For phenol and guaiacol ECH, Pt, Pt/ACC (activated carbon cloth), Ru/ACC and Raney nickel have been widely reported to be effective for the ECH reactions of these compounds[12,16–19]. However, the effective ECH of benzoic acid remains a challenge due to increasing some undesired side reactions

[1]School of Water, Energy and Environment, Cranfield University, Cranfield, MK43 0AL, UK. [2]Renewable and Sustainable Energy Research Centre, Technology Innovation Institute, Abu Dhabi, UAE. ✉e-mail: ying.jiang@tii.ae; j.sumner@cranfield.ac.uk

**Fig. 1 | Schematic representation of the electrochemical cell.** The cathode surface represents the deposited platinum (in white) and ruthenium (in green) nanoparticles.

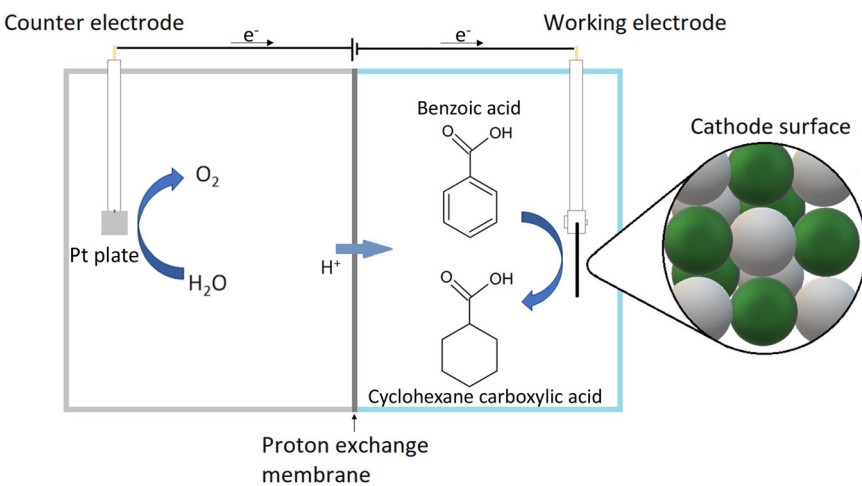

## Results and Discussion

(carboxyhydrogenation and over-hydrogenation) alongside the hydrogenation of the aromatic ring[20]. A 'bimetal' Pt/Ru catalytic system has been reported by Du et al.[20] and Fukazawa et al.[21] for benzoic acid ECH, which performed well in promoting both hydrogenation and direct hydrogenolysis. Salakhum et al.[22] found that combining both metals facilitates the cleavage of C-O and O-H bonds, reducing their heterolytic activation energies, which increases ECH effectiveness.

Understanding the complex interplay of model compounds during electrocatalytic hydrogenation (ECH) is crucial for translating this process into industrial applications when dealing with real bio-oil feedstock. The presence of multiple components can significantly influence hydrogenation pathways, leading to synergistic and antagonistic effects. For instance, competition for adsorption sites can alter product distribution. Consequently, an in-depth understanding of co-hydrogenation phenomena is essential to optimise ECH processes for industrial relevance.

This study investigates the mild hydrogenation of bio-oil model compounds using a PtRu/ACC catalyst. To elucidate the complex reaction pathways and kinetics during mixtures, we adopted a full factorial experimental design to examine the co-hydrogenation of benzoic acid, phenol and guaiacol. Additionally, the ECH mechanisms of model compounds were analysed, including side reactions, adsorption energy, and molecule placement, using in situ electrochemical measurements and density functional theory (DFT) calculations.

## Results and Discussion
### Electrochemical hydronation of model compounds

Benzoic acid electrochemical hydrogenation tests were carried out using a H-type electrochemical cell (Fig. 1). At an initial BA concentration of 10 mM, 94.70% conversion with 100% selectivity to cyclohexane carboxylic acid (CCA) was achieved within 4 h at 9% Faradaic efficiency (Fig. 2a, b). This high selectivity was consistently observed across all benzoic acid experiments in this study. Increasing the initial concentration from 10 to 20 mM resulted in a decreased conversion rate and increased FE (75.4% and 14%, respectively). The initial BA concentration significantly impacted both conversion rate and Faradaic efficiency (FE). Increasing the concentration from 10 to 20 mM enhanced both parameters, likely due to increased collision frequency between $H^+$ ions and organic molecules. However, further concentration increases to 30 and 50 mM led to decreased conversion rates, despite slightly improved FEs. This suggests a mass transport limitation at higher concentrations, where the availability of electrons becomes insufficient to sustain the reaction rate. Based on this finding, the optimal initial BA concentration for this study was determined to be 20 mM and applied to other experiments across this study.

Temperature significantly influenced ECH performance. To evaluate the effect of temperature on the process, and determine the optimal condition, we tested temperatures of 40, 50, 55, 60 and 80 °C (Fig. 2c, d), maintaining the same current density at 100.0 mA cm$^{-2}$ and setting the initial concentration of BA to 20 mM. At lower temperatures (40 °C), the system's conductivity was reduced, leading to insufficient activation energy for bond cleavage[10,12,23] and consequently lower conversion (49.8%) and Faradaic efficiency (9%). Increasing the temperature to 50–60 °C enhanced both conversion and FE, with optimal performance observed at 55 °C. However, further heating to 80 °C promoted hydrogen desorption from the cathode, resulting in decreased ECH activity (78.74% conversion, 14% FE).

The influence of current density on ECH was investigated using five current values (0.1, 0.2, 0.4, and 0.6 A), corresponding to current densities of 12.5, 25.0, 50.0, 100.0, and 150.0 mA cm$^{-2}$, respectively (based on a 4.0 cm$^2$ cathode area). The initial BA concentration was maintained at 20 mM, and the temperature at 55 °C. Preliminary tests at 0.05 A indicated insufficient proton generation on the cathode surface, limiting reactant collisions and hindering process efficiency.

Increasing the current density from 12.5 to 25.0 mA cm$^{-2}$ significantly improved both conversion rates and Faradaic efficiency. At a current density of 50.0 mA cm$^{-2}$, the conversion rate increased by 8.87%, but the FE reduced significantly from 56% to 31%. At higher current densities of 100 and 150 mA cm$^{-2}$, higher overpotentials are required (0.340 and 0.416 V vs Ag/AgCl, respectively), increasing HER rate and resulting in the lowest FE values observed in the experiment, at 14% and 10%, respectively. This decline in Faradaic efficiency at higher overpotentials can be attributed to several factors. First, higher overpotentials increase the likelihood of side reactions, which in this process is hydrogen evolution reaction (HER), these compete with the desired reaction and consume part of the applied current without contributing to the target product formation, reducing FE. Additionally, the rapid generation of hydrogen gas at elevated current densities can lead to the accumulation of gas bubbles on the cathode surface, which impedes effective mass transport by blocking active sites, thereby reducing the available surface area for the reaction[10].

To assess cathode stability, reproducibility, and process sustainability, the optimised conditions (temperature = 55 °C, current density = 25.0 mA cm$^{-2}$, initial BA concentration = 20 mM) were replicated over five consecutive cycles. The results (Fig. 3) indicated a consistent performance of the PtRu/ACC catalyst over the five cycles of the experiment. Despite the aggressive reaction environment, there is no sign of cathode degradation, which was confirmed by scanning electron microscopy observation in addition to assessing its consistency in performance via GC-MS. It is worth noting that the selectivity for CCA remains 100% throughout all cycles.

Phenol and guaiacol were subjected to ECH under the optimised conditions determined for benzoic acid. Phenol was observed to be the most effective under these conditions, achieving 83.17% conversion and 60% Faradaic efficiency with high selectivity (99%) towards cyclohexanol

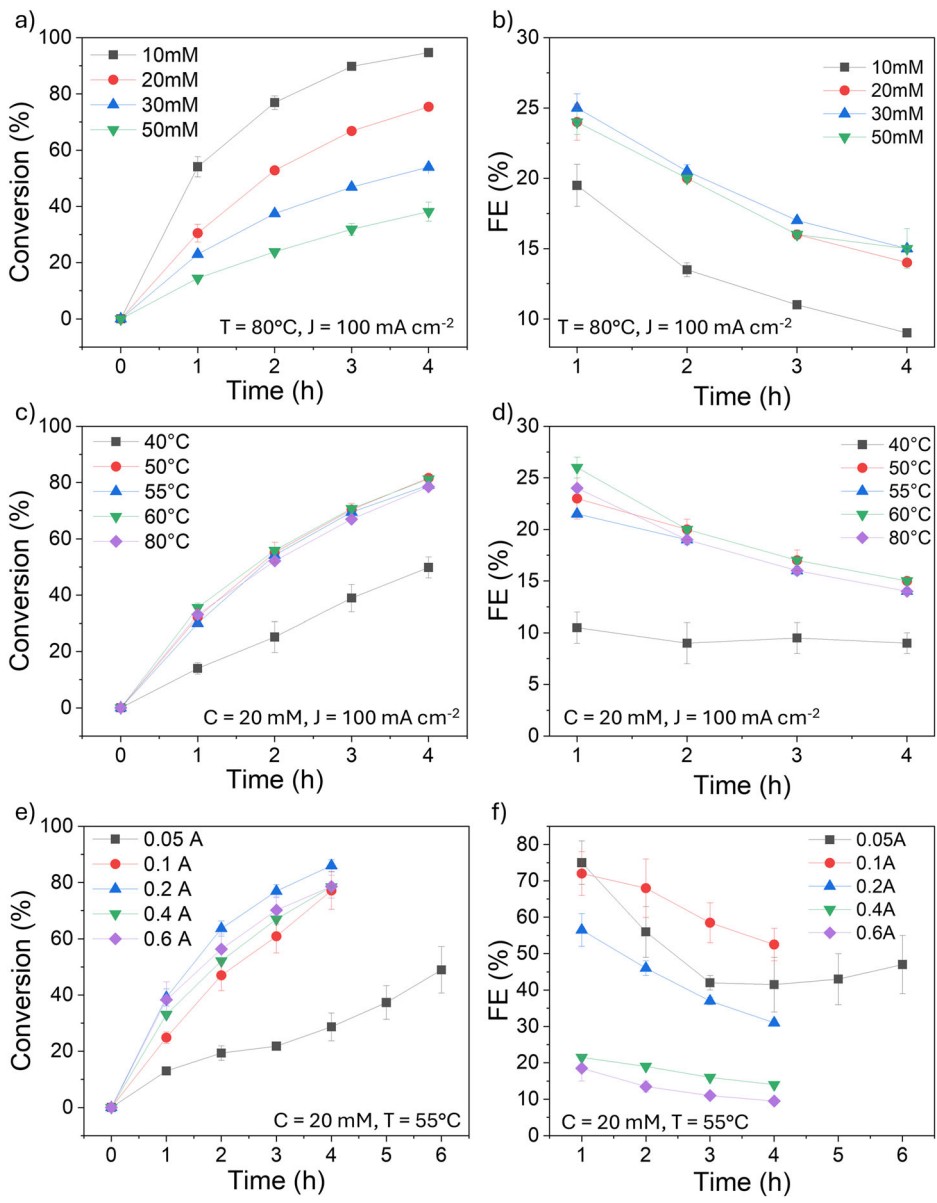

**Fig. 2 | Benzoic acid conversion to CCA over time during ECH.** Charts show the impact of different initial benzoic acid concentrations (**a**), temperatures (**c**) and currents (**e**) and their corresponding Faradaic efficiency (**b**, **d** and **f**). Where T = temperature, J = current density and C = initial concentration. Error bars indicate the standard error of triplicated experimental results.

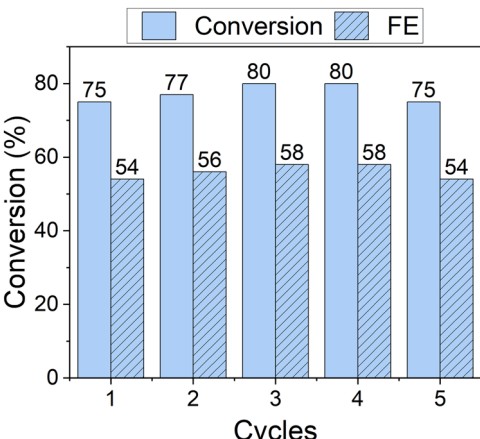

**Fig. 3 | ECH of benzoic acid in the optimal conditions for several cycles.** Temperature = 55 °C, current density = 25.0 mA cm$^{-2}$ and initial concentration = 20 mM of BA.

(Fig. 4a, b). In contrast, guaiacol demonstrated lower activity (68.59% conversion, 59% FE) (Fig. 4c, d) with a product distribution of 50% cyclohexanol, 46% 2-methoxycyclohexanol, and 4% 1-methoxycyclohexane, indicating a preference for demethoxylation over dehydration. Notably, complete aromatic ring hydrogenation was observed for all three model compounds, suggesting a common reaction pathway.

While higher conversion rates and FE for benzoic acid ECH have been reported in the literature using smaller H-type cells (25 mL vs. 90 mL)[20], direct comparison is challenging due to differences in cell geometry. Smaller cells often exhibit lower resistance and require lower overpotentials, which in turn affects FE. Scaling up to larger H-type cells necessitates increased reactant quantities, higher currents, and overpotentials, potentially impacting efficiency and selectivity. To address these limitations, continuous flow reactors may offer advantages by mitigating mass transfer issues and enabling better process control.

## Electrochemical hydrogenation of model compound mixtures
Understanding the effects of having more than one model compound during ECH is essential to progress bio-oil upgrading, which are inherently mixed compounds. To investigate potential interactions (synergistic and

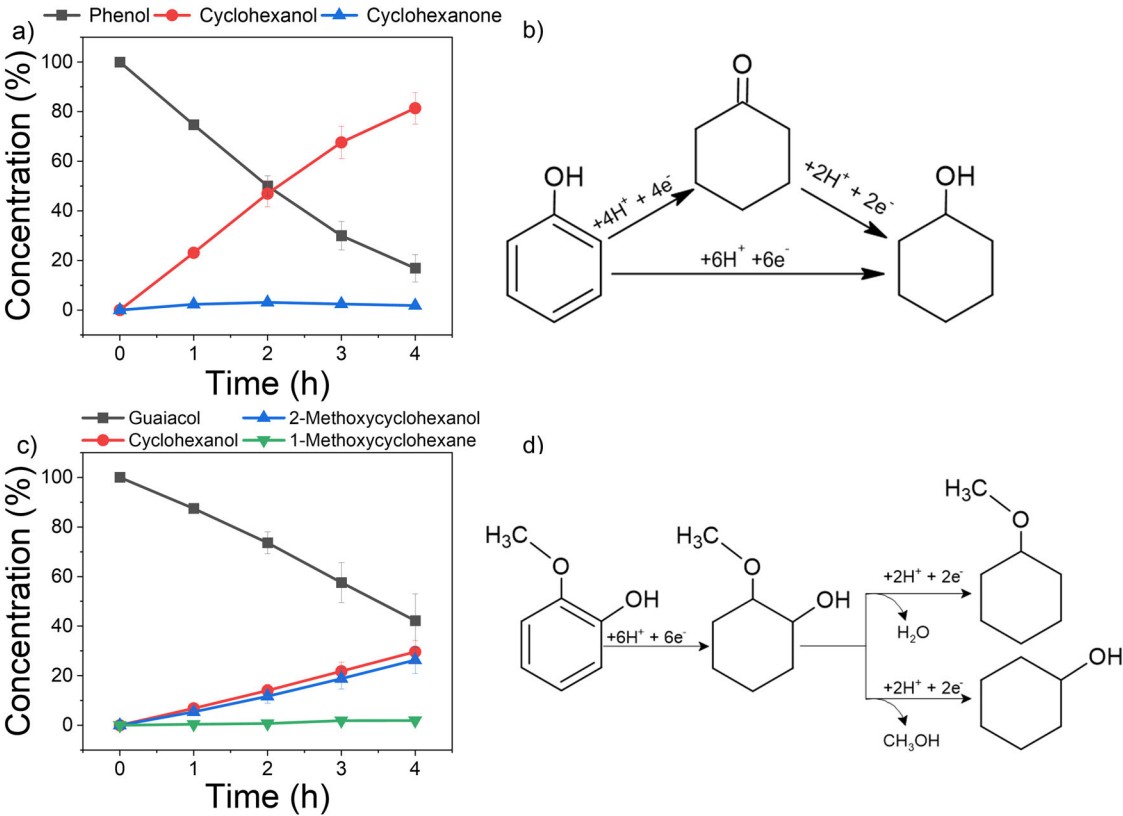

**Fig. 4 | ECH over time of guaiacol and phenol.** Conversion of guaiacol (**a**) and phenol (**c**) and their respective possible hydrogenation paths (**b** and **d**). Error bars indicate the standard error of triplicated experimental results.

**Table 1 | Results comparison for the ECH of benzoic acid (BA), phenol (P) and guaiacol (G) mixtures**

| Compound | | | Entry | Initial concentration (mM) | Conversion (%) | FE (%) |
|---|---|---|---|---|---|---|
| BA | P | G | | | | |
| + | - | - | (1) | 20 | 77.35 | 56 |
| - | + | - | (2) | 20 | 88.75 | 64 |
| - | - | + | (3) | 20 | 68.59 | 59 |
| + | + | - | (4) | 40 | 64.19 | 74 |
| + | - | + | (5) | 40 | 57.27 | 66 |
| - | + | + | (6) | 40 | 47.80 | 55 |
| + | + | + | (7) | 60 | 42.59 | 73 |
| + | + | - | (8) | 20 | 80.02 | 58 |
| + | - | + | (9) | 20 | 63.01 | 46 |
| - | + | + | (10) | 20 | 74.78 | 54 |
| + | + | + | (11) | 20 | 33.64 | 24 |

Reaction parameters: current density = 25.0 mA cm$^{-2}$, temperature = 55 °C, electrolyte = H$_2$SO$_4$ 1 M.

**Table 2 | Benzoic acid ECH optimisation tests and ECH of both phenol and guaiacol at the optimised conditions**

| Compound | Concentration (mM) | T (°C) | Current (A) |
|---|---|---|---|
| Benzoic acid | 10 | 75 | 0.4 |
| Benzoic acid | 20 | 75 | 0.4 |
| Benzoic acid | 30 | 75 | 0.4 |
| Benzoic acid | 50 | 75 | 0.4 |
| Benzoic acid | 20 | 40 | 0.4 |
| Benzoic acid | 20 | 50 | 0.4 |
| Benzoic acid | 20 | 55 | 0.4 |
| Benzoic acid | 20 | 60 | 0.4 |
| Benzoic acid | 20 | 80 | 0.4 |
| Benzoic acid | 20 | 55 | 0.05 |
| Benzoic acid | 20 | 55 | 0.1 |
| Benzoic acid | 20 | 55 | 0.2 |
| Benzoic acid | 20 | 55 | 0.6 |
| Phenol | 20 | 55 | 0.1 |
| Guaiacol | 20 | 55 | 0.1 |

antagonistic effects) of each compound in the mixture, a factorial experimental design approach was applied as shown in Table 1, 2, where a "+" signs indicate the compound is present, and a "-" signals the opposite. Notably, the applied overpotential showed minimal variance (0.003 V), remaining between 0.205 and 0.208 V vs Ag/AgCl, regardless of the composition or concentration of the model compounds. This indicates that the conductivity of the system is more significantly influenced by the electrolyte and temperature rather than the model compounds, likely due to their low concentration.

It was observed that increasing the number of unsaturated molecules generally decreased conversion rates, with the exception of mixtures containing all three compounds (Entries 7 and 11). Conversely, FE increased in runs 4, 5 and 7. Figure 2a shows a clear correlation between decreasing conversion rates and increasing initial concentration.

Notably, in entry 4 a significant drop in conversion from 83.05% (average of BA and P) to 64.19% in the mixture was observed, accompanied by a 13% increase in FE. This suggests a possible synergistic interaction

**Fig. 5 | Proposed hydrogenation path for guaiacol and phenol in mixtures.** Where the formation of phenol was observed and the full hydrogenation to cyclohexane was achieved.

between these two compounds, which aligns with previous studies on phenol-containing mixtures (phenol + furfural[24], and phenol + benzaldehyde[25,26]), where hydrogen-bonded complex formation enhanced the hydrogenation of the co-reactant.

Phenol exhibited the highest conversion rates among the model compounds. In the BA + P mixture (Entry 4), phenol conversion reached 88.57% compared to 39.81% for benzoic acid. Similarly, phenol was preferentially converted in the G + P mixture (Entry 6) at 74.16% compared to 21.43% for guaiacol. This trend persisted in the ternary mixture (Entry 7), with phenol conversion at 75.79% exceeding those of benzoic acid (18.94%) and guaiacol (33.04%). Phenol also demonstrated the highest overall conversion efficiency as a standalone compound.

As a standalone compound, guaiacol consistently exhibited lower conversion rates compared to phenol and benzoic acid. This reduced reactivity is attributed to its complex molecular structure containing both methoxy and phenolic groups, resulting in higher bond energies[19]. Interestingly, in mixtures, the hydrogenation of guaiacol appears to be favoured when compared to BA. Mixing benzoic acid with guaiacol (entry 5), led to 50% and 65% conversion of each compound, respectively. This value is quite similar to pure guaiacol tests (69%, entry 3), despite the higher initial concentration. This trend was also observed for entry 7 (mixture of all three compounds).

The ECH of mixtures also led to different products, including, although in small quantities ( < 2.0 area %), the full hydro-dehydroxylation of single aromatic compounds into cyclohexane. Guaiacol was also converted into phenol and cyclohexanone when mixed with BA, which complicates the calculations for the conversion rates of phenol. Where, if this behaviour is duplicated in other mixtures, it becomes increasingly difficult to precisely calculate to which extent the present molecules are the ones added into the system as a model compound or produced via ECH. Therefore, the conversion and FE of said processes could be higher than the reported values. The proposed hydrogenation path for guaiacol and phenol in mixtures is shown in Fig. 5.

Tests were carried out to observe the impact of the initial concentration on mixtures. These followed the same ratio between compounds, but the initial total concentration of compounds for ECH was set to 20 mM. For entries with two compounds, each compound contributed 10 mM, and for entries with three compounds, each compound contributed 6.7 mM.

Setting the initial concentration to 20 mM led to similar trends to the ones observed previously. The BA + P mixture (entry 8) remains the best result for a mixture, presenting a considerably high 69.70% conversion for BA and 90.34% for phenol, very similar results to entries 1 and 2, despite the reduction in the initial concentration for each compound (from 20 to 10 mM), once again remarking to possible synergistic effects between these

two compounds. Interestingly, although a slight decrease in activity is to be expected, this did not occur for phenol. Phenol also remained the most active species for ECH, achieving 91.44% conversion, compared to 58.12% for guaiacol (entry 10) and 61.60% compared to 24.33% and 26.11% for BA and guaiacol (entry 11), respectively.

Although the mixtures with guaiacol still presented inferior results, it is worth noting that P + G at 20 mM (entry 10) is far superior to P + G at 40 mM (entry 6), possibly due to guaiacol's resistance to ECH. That is, increasing guaiacol's concentration meant a greater negative interference in the process than phenol's positive one, leading to a lesser result. No phenol was detected in entry 9 (BA + G), and there was also no cyclohexane found in any of the experiments. This suggests that the concentration of these compounds may have been too low to detect, or that a larger quantity of reactant was needed to achieve the full conversion. The full GC-MS spectrum of all the experiments can be found at the Supplementary information (Supplementary Figs. 6–15).

**Factorial design analysis**

Two models were developed to study the effects of mixture composition at low and high concentrations. In both cases, the initial model included main effects and interaction terms: BA, P, G, BA + P, BA + G, and P + G, with conversion and FE as responses.

For conversion at higher concentrations (Fig. 6a), significant contributions were observed from BA, BA × P, BA × G, and P × G, while P and G were excluded from the final model. The $r^2$ value for conversion was 0.7267, indicating that the model explains a moderate proportion of the variability in the response. For FE (Fig. 6b), all terms significantly contributed, with an $r^2$ value of 0.9290, indicating a strong model fit.

At lower concentrations, G was excluded from the final model for conversion, while BA and G were removed for FE. The $r^2$ value for conversion was 0.9608 (Fig. 6d), and for FE, $r^2$ was 0.9325 (Fig. 6e), both demonstrating a high correlation with the predicted model.

The conversion profiler (Figs. 6c, f) shows that increasing the concentration of the mixture generally leads to higher FE, with one exception. Phenol had the greatest positive impact on both conversion and FE, regardless of concentration. Conversely, Guaiacol decreased both responses in 3 out of 4 cases but led to a slight increase in FE at higher concentrations. These findings reaffirm the observed effects of these two compounds in the ECH process.

Residual plots for both models' responses showed that the residuals appeared random, with no discernible patterns or trends, confirming the model's assumptions. Additionally, no residuals exceeded control limits, indicating a good model fit. The plots of actual versus predicted responses showed a high correlation, confirming the model's predictive capability.

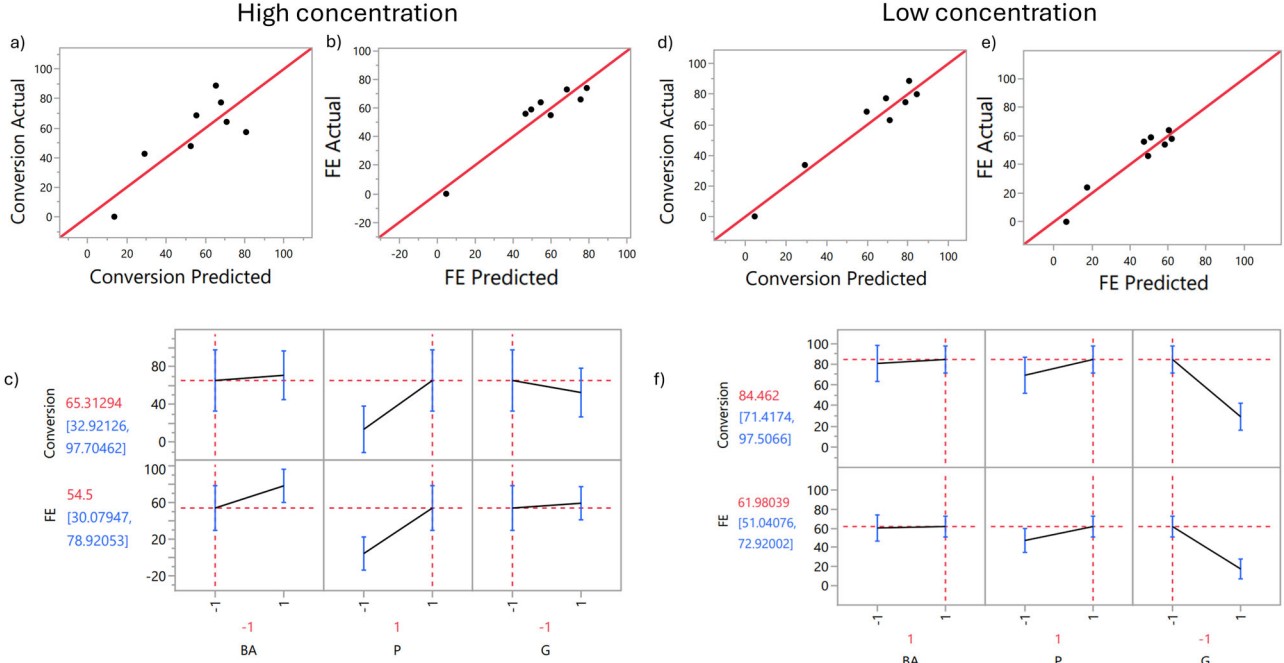

**Fig. 6 | Factorial design analysis for lower and higher concentrations.** Best regression fit for the models of predicted vs real conversion (**a,d**) and real FE (**b,e**) and conversion profiler for all model compounds (**c,f**).

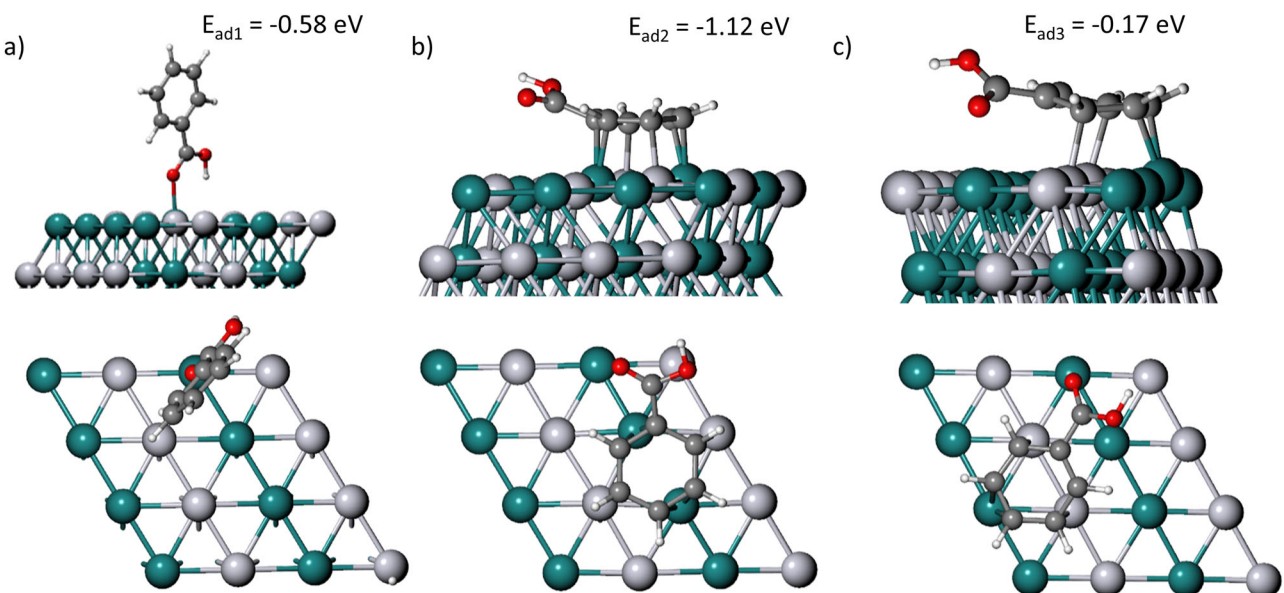

**Fig. 7 | Vertical and parallel adsorption of benzoic acid into PtRu(111). (a)** Vertical placement and (**b**) horizontal placement into PtRu(111). (**c**) Adsorption into a different site. The respective top-view of the first layer is shown in the bottom images. Where the white spheres represent hydrogen, dark grey carbon, red oxygen, light grey platinum and green represent ruthenium atoms.

## Density Functional Theory (DFT) calculations

DFT calculations were used to explain further the behaviour of benzoic acid on the cathode surface. The material was built with a Pt-Ru atomic ratio of 1:1, and the selected plane was (111), which was chosen to maintain cohesion with the reference utilised[20], the slab will be herein denoted PtRu(111). BA was placed both with a parallel placement and at a 45° angle into the surface and the structure was relaxed to find the adsorption energy. Figure 7 shows the interaction between them (See Supplementary Data 1).

The lower energy value calculated for the parallel placement gives a good interpretation of the 100% selectivity of BA into CCA, whereas there is no bond between the carboxylic acid part of the molecule and the surface.

The atom's placement shows that this group moves in the "z" direction, bending the molecule and moving away from the surface. That is, there is the formation of π-σ interactions between the aromatic ring and the metal surface, rather than the carboxylic group. There was also a difference in $E_{ad}$ depending on the initial position of the compound, where placing the aromatic ring closer to ruthenium atoms (Fig. 7b) led to a more stable system ($E_{ad2} = -1.12$ eV) than closer to platinum atoms ($E_{ad3} = -0.17$ eV, Fig. 7c), signifying it as the preferential site.

Interestingly, Fig. 7b shows that a defect is created in the first layer of the cathode's surface. To accommodate the BA molecule, the two ruthenium atoms, which bonded with C1, C3, C4 and C6 moved too far from

each other, severing the bond. To determine whether this was a one-time event, the atoms were placed closer to each other until a bond was formed, but the relaxation process would always lead to this defect. It is worth mentioning that from the second layer onwards, this did not occur.

The electron density difference (Fig. 8) shows an electron accumulation (in yellow) in the interface between the adsorbed molecule and the surface, corresponding to the adsorption bonds. In contrast, the blue area shows a clear electron depletion in the aromatic ring. The carbonyl group, an electron-accepting group, pulls the electron density towards itself, degrading the electron density of the phenyl group, an electron-donating group. This creates a positively charged area which strongly adsorbs into the surface. Further accumulation of electrons around ruthenium atoms facilitates the hydrogenation reaction.

It is widely acknowledged that the Langmuir-Hinshelwood (L-H) adsorption mechanism is followed by organic compounds containing an aromatic ring during ECH[20]. The process involves $H^+$ being quickly adsorbed into the surface, due to the synergistic effects between Pt and Ru where, in combination with an $e^-$, produces $H_{ads}$ (Volmer step). The next step is the adsorption of BA in a position parallel to the catalyst surface, to form $BA_{ads}$. Subsequently, $H_{ads}$ is transferred into $BA_{ads}$ until the full hydrogenation of the aromatic ring is accomplished, followed by the

desorption of the product (cyclohexane carboxylic acid). The proposed mechanism for the ECH of benzoic acid over PtRu/ACC is shown in Fig. 9.

The hydrogenation happens in a specific order, following the most stable state of the molecule. Calculations on the one-by-one hydrogenation (Supplementary Fig. 1) of BA show that the first carbon to receive a $H^+$ is C1 (meta), followed by the order C2-C3-C4-C5-C6. Due to the carboxylic acid's electronegativity, a negatively charged dipole ($\delta^-$) is formed which is close to C6, giving a good justification for why it is the least stable position for the first hydrogen (H1), and why it is the last to be hydrogenated. The positive dipole on C6 (ortho), which borders C1 and C5 (meta) and the distance from C3 to the -COOH group set them as the major candidates for H1, at a relatively small 0.056 and 0.0147 eV difference from C1 to C5 and C3, respectively. The same trend follows for the rest of the process, where the neighbouring carbon to the one that receives the hydrogen is always the least stable position until the ring is fully stabilised.

## Material characterisation

Figure 10 shows the Scanning Electron Microscopy (SEM) secondary electron image for PtRu/ACC before use and after 5 cycles of the optimised BA electrochemical hydrogenation process. In both samples, the metal deposition is homogeneous throughout the surface of the carbon cloth. Elemental mapping (by energy dispersive X-ray spectroscopy) was carried out and revealed an even distribution of both Pt (in teal) and Ru (in green) across the surface; this even distribution is beneficial to optimise the reaction. No degradation of the cathode is apparent after 5 reruns, the concentration and distribution of metals remain unaltered.

XRD data (Fig. 10g) shows peaks corresponding to carbon, platinum, and a 1:1 mixture of platinum and ruthenium, consistent with the expected results. The peaks are broad and exhibit low intensity, indicating the presence of amorphous content, which complicates the interpretation of the results. The sharp peak at 18 degrees is attributed to PTFE (polytetrafluoroethylene), used in various steps of the manufacturing process of the carbon cloth, including the base cloth and the micro-porous layer (MPL). Raman spectroscopy (Fig. 10h) presents the spectra of the cathode before (in black) and after (in red) the addition of PtRu to the carbon. The black spectrum displays two peaks around 1350 cm$^{-1}$ and 1580 cm$^{-1}$, which correspond to carbon. In contrast, the red spectrum shows no apparent peaks,

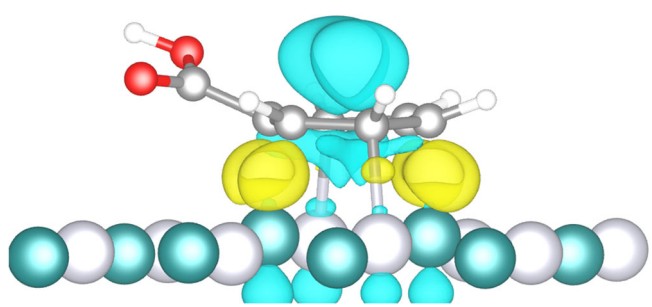

**Fig. 8 | Energy density difference of benzoic acid adsorbed into PtRu(111).** Electron depletion (light blue) and electron accumulation (yellow) isosurfaces are set to 0.06 e/Å³. Where the white spheres represent hydrogen, dark grey carbon, red oxygen, light grey platinum and green represent ruthenium atoms.

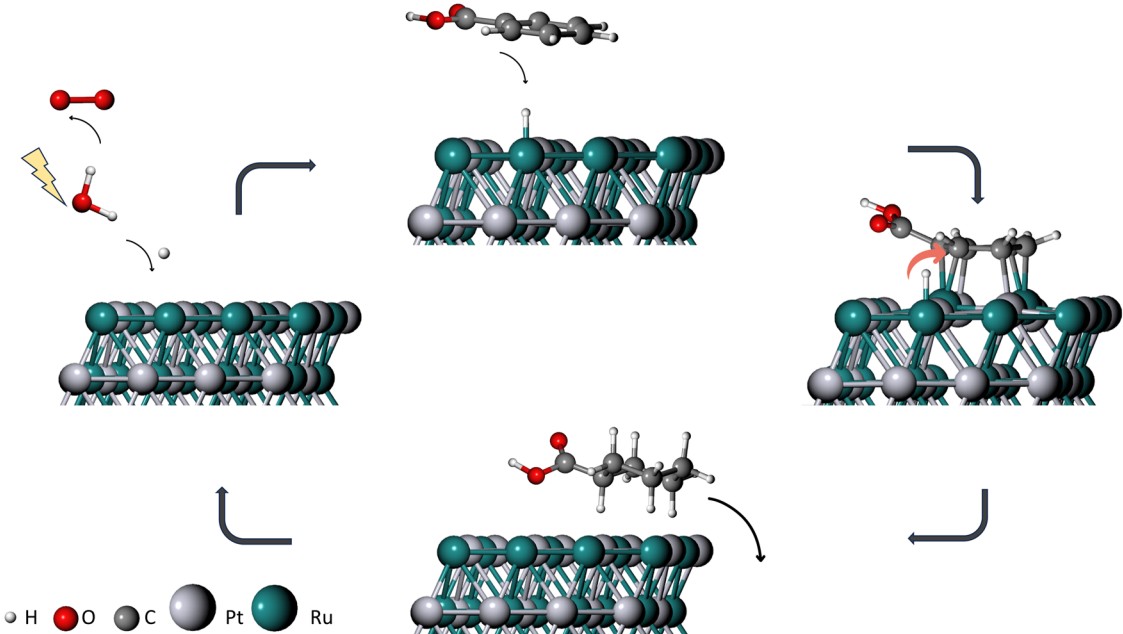

**Fig. 9 | Proposed mechanism for benzoic acid hydrogenation over PtRu/ACC, where the hydrogenation process of BA is carried out following the Langmuir-Hinshelwood mechanism for the formation of CCA.

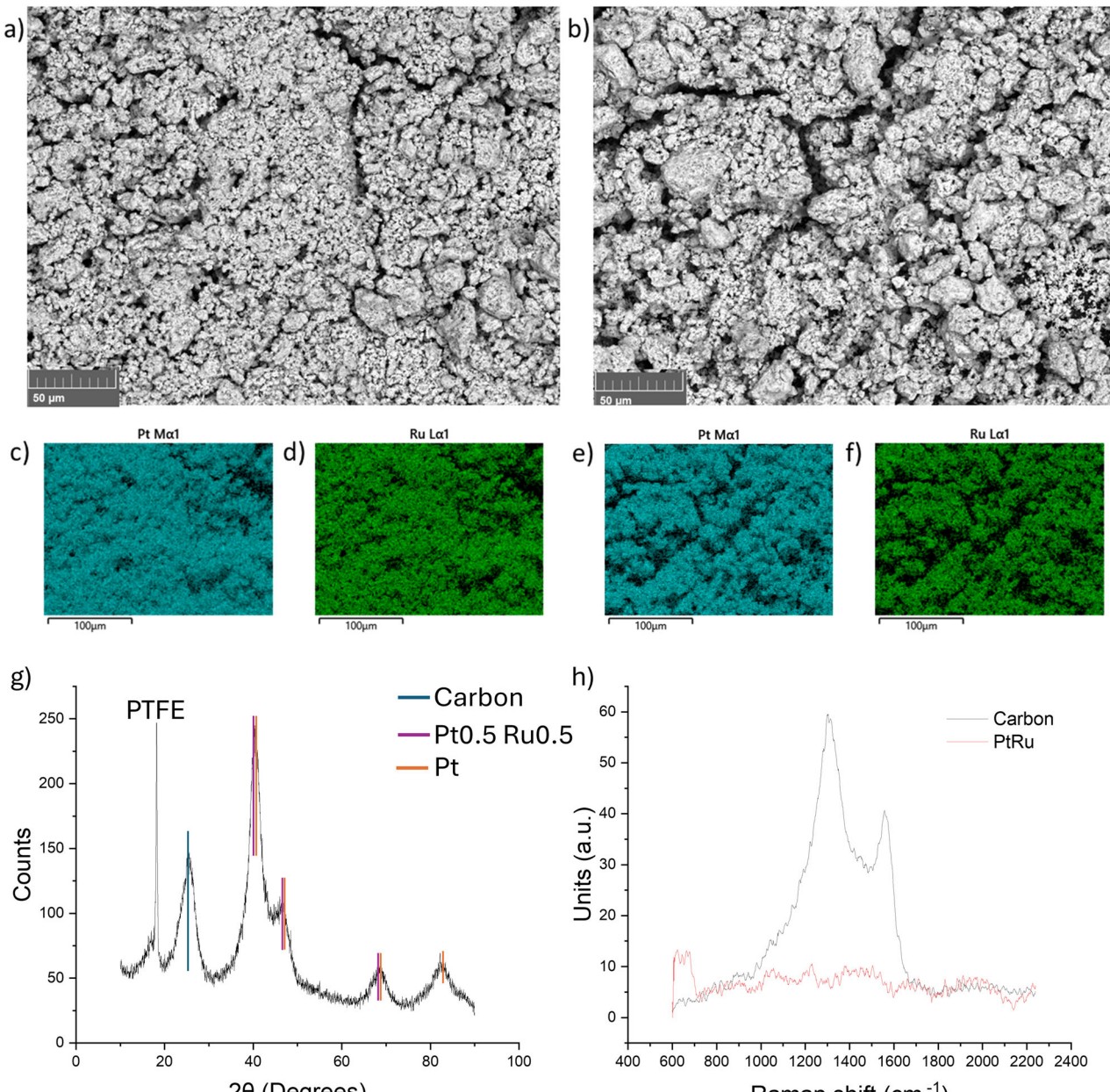

**Fig. 10 | Material characterisation via SEM, XRD and Raman.** Scanning Electron Microscopy of the cathode before (**a**) and after 5 runs (**b**). Elemental mapping before (**c** and **d**) and after 5 runs (**e** and **f**). Pt was highlighted in teal and Ru in green. **g** XRD analysis, where the peaks for carbon, platinum and PtRu are highlighted in teal, purple and orange respectively. **h** Raman spectroscopy of the cathode with (in red) and without (in black) PtRu deposition.

as both pure platinum and pure ruthenium typically do not exhibit significant Raman signals.

The cyclic voltammetry (Fig. 11a) data reveal that the addition of benzoic acid (in black), phenol (in red), and guaiacol (in green) to the PtRu/ACC electrode system in sulfuric acid progressively decreases the current response compared to the baseline (in yellow) established with only the electrolyte present. This trend suggests that each compound interacts differently with the electrode surface, likely through adsorption or electron-transfer reactions that compete with or inhibit the baseline electrochemical processes involving protons, such as hydrogen evolution reaction (HER). The yellow curve, which represents the system with only the electrolyte, shows the highest peak current, reflecting the uninhibited activity of the PtRu/ACC electrode in an acidic environment. However, with the introduction of the model compounds, the current response decreases. Phenol, while more electroactive, results in a substantial drop in current, indicating it

may strongly interact with the electrode, likely due to its higher propensity for reduction. Conversely, guaiacol, exhibits the lowest current, suggesting the strongest surface adsorption or passivation effect among the compounds tested. The asymmetry of the CV curves, particularly the lack of mirrored oxidation and reduction peaks, indicates that the electrochemical processes in this system are of nonreversible nature[27,28].

LSV analysis shows that PtRu/ACC exhibits -0.194 V vs Ag/AgCl at 10 mA cm$^{-2}$, and it becomes less negative when adding the compound, reaching -0.163 V, -0.162 V and -0.159 V vs Ag/AgCl for G, BA and P, respectively, (Fig. 11b, c). This measurement is important for the well-known activity for HER at this current density and it is also an indication that the reactivity of the compounds for ECH is P > BA > G, which is corroborated by the empirical data[28].

The voltage change ($\Delta E$) of $\Delta E_1 = 0.031$ V, $\Delta E_2 = 0.032$ V and $\Delta E_3 = 0.035$ V shows that the ECH of all three model compounds are prior

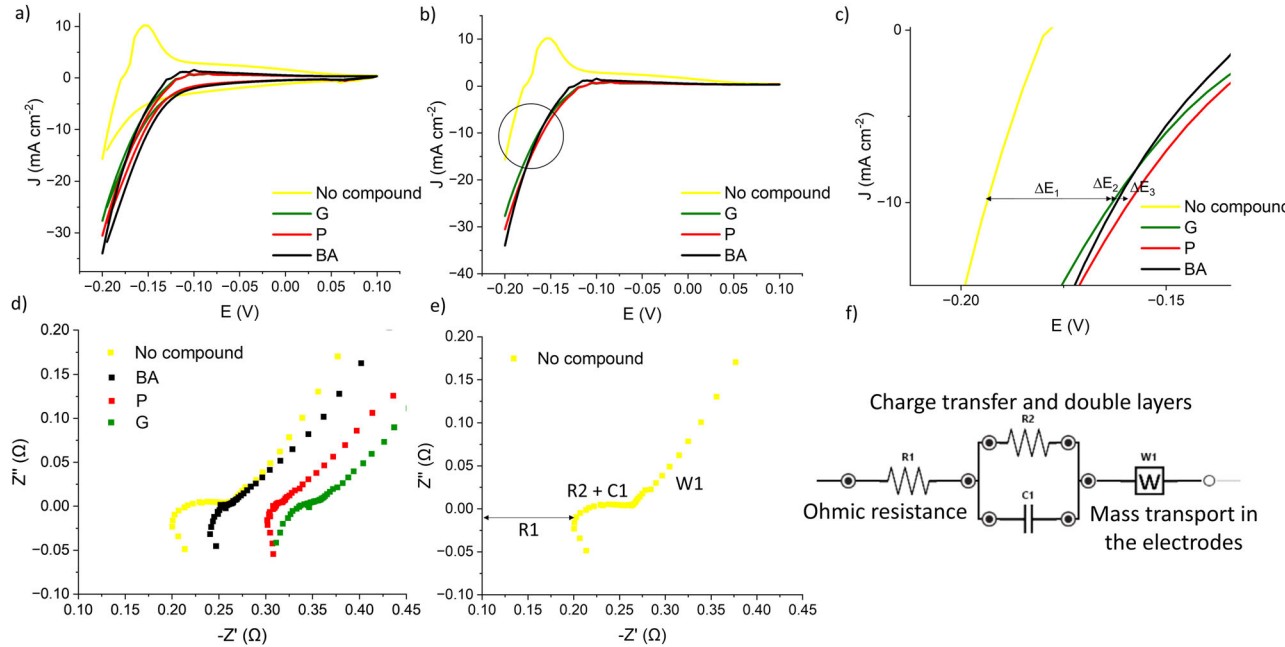

**Fig. 11 | Electrochemical analysis.** Cyclic voltammetry (**a**) of PtRu/ACC before (in yellow) and after the addition of benzoic acid (in black), guaiacol (in green) or phenol (in red) into the system (1.0 M H$_2$SO$_4$, 55 °C). Linear sweep voltammetry (**b**) and a zoomed-in version (**c**). Where ΔE$_1$, ΔE$_2$ and ΔE$_3$ are the differences between no added compound and G (guaiacol), BA (benzoic acid) and P (phenol) at 10 mAcm$^{-2}$, respectively. (**d**) EIS and zoomed-in version (**e**) highlighting the baseline curve. **f** Equivalent circuit for EIS fitting. All measurements are vs Ag/AgCl reference electrode.

to HER, the competing reaction, conversely. When adding P to BA, the potential went from -0.162 V to -0.155 V vs Ag/AgCl, a difference of 0.07 V, whereas when adding G to BA, there was a negative shift, going from -0.163 to -0.185 V indicating an anti-synergistic effect. Adding P to G increased the potential from -0.163 to -0.152 V, a 0.11 V change (see Supplementary Figs. S2–4).

In the Nyquist plot (Fig. 11d), the baseline (yellow curve) represents the system before the addition of model compounds, while the black, green and red curves correspond to the presence of benzoic acid, guaiacol and phenol in the electrolyte, respectively. The EIS data reveal an increase in the ohmic resistance and the diameter of the semicircle upon the addition of the model compounds, indicating a higher charge transfer resistance (R$_{ct}$) in the presence of compounds. This rise in R$_{ct}$ suggests that the hydrogenation requires additional energy input, likely due to the formation of surface-bound intermediates that create an energy barrier to electron transfer. The curves also show a tail inclined at a 45° angle, which is characteristic of the Warburg impedance semi-infinite diffusion layer and indicates diffusion limitations[29]. These diffusion limitations likely arise from mass transport effects, such as the diffusion of hydrogen or model compound molecules to and from the electrode surface.

## Conclusions

In conclusion, the electrochemical hydrogenation of mixtures represents a significant advancement in the field of bio-oil upgrading. Through a factorial experimental design approach, we explored the interactions among model compounds, shedding light on both synergistic and antagonistic effects within the system. Our findings underscored the robustness of the electrochemical process, with the applied overpotential exhibiting minimal variance across different compositions and concentrations of model compounds. Notably, we observed a clear trend where increasing the number of unsaturated molecules in the system led to reduced conversion rates, except when all three compounds were mixed, suggesting complex dynamics at play. Phenol emerged as the most readily converted compound in most scenarios, highlighting its preferential hydrogenation over benzoic acid and guaiacol. The resistance to guaiacol hydrogenation, attributed to its complex molecular structure with methoxy and phenolic groups, posed challenges in

achieving high conversion rates, particularly when guaiacol was present in mixtures. However, the hydrogenation of guaiacol was favoured over benzoic acid, indicating nuanced reactivity patterns influenced by compound interactions.

Furthermore, our investigation into the impact of initial concentration revealed consistent trends, with mixtures maintaining similar conversion patterns despite variations in compound concentrations. Phenol consistently demonstrated higher activity compared to benzoic acid and guaiacol, even at reduced concentrations, underscoring its significance in the electrochemical hydrogenation process.

DFT calculations demonstrated that positioning the BA molecule parallel (0°) to the surface was notably more stable compared to a 45° angle placement. This preference suggests a stronger affinity for the adsorption of the aromatic ring over the carboxylic acid component of the molecule. Such findings provide insight into the observed 100% selectivity of product formation towards cyclohexane carboxylic acid.

Overall, our study provides valuable insights into the electrochemical hydrogenation of mixtures, offering a comprehensive understanding of compound interactions and their influence on conversion rates and selectivity. These findings lay the groundwork for further optimisation of electrochemical processes in bio-oil upgrading, with potential applications in sustainable energy production and chemical synthesis.

## Methods
### Materials

Sulphuric acid solution (5 M), benzoic acid (99%, extra pure) and guaiacol (99%) were purchased from Thermo Fisher Scientific. Phenol (liquified ≥89%) was purchased from Sigma Aldrich. All chemicals were used as received without further purification. A Nafion 117 membrane (Ion Power Inc. PA, USA) was used as the cation exchange membrane to separate the cathode and anode chambers. To ensure the activation of the membrane and optimise its ion transport capability, it was pretreated according to Peng et al. [30]. It was treated in oxygen peroxide (H$_2$O$_2$) at 80 °C for 1 h, followed by deionised water for 1 h and 1.0 M H$_2$SO$_4$ for 1 h at the same temperature. The membrane was washed with deionised water after each stage. The utilised cathode is a 2x2cm, 4 mg/cm² Platinum Ruthenium Black-Carbon

Cloth Electrode (PtRu/ACC, Fuel Cell Store Inc., Texas, USA, 1:1 Pt:Ru ratio).

## ECH of pure and mixed model compounds

A two-chamber electrochemical cell, separated by a proton exchange membrane (PEM) was selected for this process, where the anode was a 1×1 cm platinum plate, the selected catalytic cathode was PtRu/ACC. Both cells were filled with 1.0 M sulfuric acid ($H_2SO_4$) as the electrolyte. The ECH tests were performed at a 90 mL (per chamber) H-type cell and run at electrostatic control and constant temperature (both at different values depending on the experiment) for 10 min without the organic compound to polarise the electrodes. Afterwards, the model compound was added to the system to obtain a set final concentration. The experiment was run for a duration which ensured the availability of sufficient electrons to convert the compounds (4 to 6 h). Samples (1 mL) were taken every hour, and the organic compounds were extracted with 2 mL of dichloromethane (DCM). After each experiment, the electrolyte was discarded, the cell cleaned and the cathode was soaked in 5 ML dichloromethane (DCM) for 15 min, to ensure the desorption of any molecules on its surface.

The process of finding the optimised conditions was done utilising benzoic acid as the main model compound, and guaiacol and phenol were used to confirm the effectivity of the process in other model compounds. All tests with mixtures were carried out at the same temperature and current density as the optimised point, variations on the initial concentration were specified in the text. All tests were carried out in triplicates and the error bars are relative to the standard error.

For analysis, the samples were tested in the GC-MS Shimadzu TQ8040. The GC used a Restek Rtx-5 capillary column, 28.5 m × 0.25 mm with a 0.25 μm film thickness, a 1.18 ml min$^{-1}$ helium carrier gas flow rate, and a split ratio of 1:25. The injector temperature was set at 250 °C. The GC oven program started at 34 °C and was held for 2 min and then heated at 50 °C min$^{-1}$ to 300 °C. Mass spectrometry was operated at m/z ranging from 50 to 500. Species associated with each chromatographic peak were identified by comparing their observed mass spectrum with the NIST library. External standards were also used to identify compounds and quantify the peaks. Results are in area %. The morphology of the electrodes and EDS (Electron Dispersive Spectroscopy) were acquired by TESCAN VEGA 3 Scanning Electron Microscope (SEM). Raman spectroscopy was performed using a Horiba LabRAM equipped with a Red 632.8 nm laser. Samples were exposed using the following parameters: no filter, 400 μm hole, 100 μm slit, 1800 grating, 3 exposures each of 5 s. Before any samples were analysed, the Raman shift was calibrated using the known 520.7 cm$^{-1}$ peak of a silicon reference sample.

## Factorial design of experiment

The experimental matrix was designed using JMP PRO 17, as shown in Table 1, where a "+" signs indicate the compound is present, and a "-" signals the opposite. Each model compound and the initial concentration were selected as independent variables (factors), and the conversion rate and Faradaic efficiency as responses. The factors were set to discrete numeric values, and the model covered all main effects and two-factor interactions. Experiments with no model compounds ("-" "-" "-") were not shown in Table 1, as no compounds would lead to 0% conversion. The initial concentration of each compound is always equal to the others. For all single-compound experiments, the duration was set to 4 h, otherwise it was 5 h. This adjustment was made to ensure enough electrons were introduced into the system, without a change in current.

## Electrochemical measurements

Electrochemical measurements were performed using a PalmSens EmStat4s at 55 °C, which consisted of a three-electrode system with the used cathode, anode, and an Ag/AgCl (in saturated KCl solution) as the working, counter, and reference electrodes, respectively. The potential vs. Ag/AgCl was converted to the potential vs.RHE (reversible hydrogen electrode) based on the Nernst equation:

$$R_{RHE} = E_{Ag/AgCl} + 0.059 \, x \, pH + E^0_{Ag/AgCl} \qquad (1)$$

All electrochemical measurements were carried out in 1.0 M $H_2SO_4$ with and without the model compounds. LSV tests were recorded at a scan rate of 5 mV s$^{-1}$ from -0.200 V to 0.200 V. CV was carried out from -0.200 V to 0.200 V, with a $E_{step}$ of 5 mV and a scan rate of 5 mV s$^{-1}$. EIS was measured vs OCP at a max frequency of $10^5$ Hz and a min frequency of $10^{-1}$ Hz, the $E_{start}$ was set to -0.200 V. Equilibration time was 5 s.

## Calculations

To understand and quantify the ECH process, the following equations were used:

$$F.E.(\%) = \frac{mol \, produced \, x \, n \, x \, F}{Total \, electrons \, passed} \, x \, 100 \qquad (2)$$

$$Conversion(\%) = \left(\frac{Moles \, reactant \, consumed}{Initial \, moles \, reactant}\right) x \, 100 \qquad (3)$$

$$Selectivity(\%) = \left(\frac{Moles \, desired \, product}{Total \, moles \, product}\right) x \, 100 \qquad (4)$$

$$Current \, density(mAcm^{-2}) = \frac{Current}{Surface \, area} \qquad (5)$$

$$Yeild(mMolh^{-1}) = \frac{Moles \, desired \, product}{Reaction \, duration} \qquad (6)$$

Where F is the Faraday constant (96 485 C mol$^{-1}$), and n is the number of electrons necessary for the hydrogenation process ($n = 6$ for the conversion of benzoic acid into cyclohexane carboxylic acid, for example).

## DFT Calculations

All calculations were carried out using QuantumEspresso/7.2-foss-2022b[31–33] with BURAI 1.3.1 as a visualiser. VESTA was used to calculate the electronic structure. Both positive and negative isosurfaces were set to 0.06 e/Å$^3$. The generalized-gradient approximation (GGA) with the Perdew-Burke-Ernzerhof (PBE) functional described the exchange-correlation energy. To enhance computational performance, all calculations were performed with a thermal smearing of 0.01 Ry and spin polarisation. A 4-Layer 4 × 4 PtRu (111) slab was built with a 20 Å vacuum gap and two fixed bottom layers. The slab was placed at the bottom of the cell, to certify that all reactions happened on the top surface. A (2 × 2 × 1) k-point grid was set. The test molecule was placed at a distance of 1.5 Å from the surface, where it did not start the calculations with any bonds with the surface. The adsorption energy (Ead) of the different adsorbates is defined as:

$$E_{ad} = E_{Total} - (E_{adsorbate} + E_{Surface}) \qquad (7)$$

Where $E_{Total}$ is the calculated energy for the final system (slab + adsorbate), $E_{adsorbate}$ is the calculated energy of each adsorbate and $E_{Surface}$ is the energy of the bare cathode surface.

## Data availability

The source data supporting this study's findings are available from the Figshare repository (https://doi.org/10.6084/m9.figshare.26540506.v1) Atom positions for DFT results are provided in Supplementary Data 1.

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

## Acknowledgements
The authors would like to thank EPSRC (EP/T518104/1) and Cranfield University for their support.

## Author contributions
C.Catizane was conducted the experimental work and analysis/imaging and was the primary author of the paper. J.Sumner and Y.Jiang supervised the research, revised the manuscript and contributed to technical content of paper.

## Competing interests
The authors declare no competing interests.
