## [Peer Review file · Communications Chemistry]

Mechanisms of electrochemical hydrogenation of aromatic compound mixtures over a bimetallic PtRu catalyst

Corresponding Author: Dr Joy Sumner

Version 0:

Reviewer comments:

Reviewer #1

(Remarks to the Author)

Efficient and selective electrochemical hydrogenation (ECH) of organic compounds is critical to reducing global dependence on fossil fuels. In this work, the authors used a PtRu/ACC catalyst for the mild hydrogenation of bio-oil model compounds. BA+P showed the highest conversion (64.19%) and faraday efficiency (74%). In addition, the selectivity of BA to cyclohexanecarboxylic acid (CCA) was consistently maintained at 100% regardless of the experimental parameters. However, the quality of the manuscript remains to be improved. Some major comments have been listed for the authors' reference to further improve the quality of this work.

1. The authors tested a variety of mixtures for conversion efficiency and faraday efficiency. It is recommended that the authors perform multiple experiments to take an average and establish error bars to further improve the accuracy of the data.
2. In this paper, there are few characterizations for the PtRu/ACC catalyst. (1) What is the atomic ratio of Pt and Ru for PtRu/ACC. This is crucial for the later modeling. (2) The XRD test is required to explore whether the material is in a pure phase or not. (3) The valence state of PtRu/ACC was characterized to further explain the catalyst mechanism. In addition to the above characterization, the authors still lack TEM, Raman, etc.
3. The authors chose the (111) plane of PtRu/ACC for calculation. Please explain why this plane was chosen instead of other crystal planes such as (110).
4. The authors only calculate the surface energy, and lack the calculation for the electronic structure of the material. It is suggested that the authors supplement the calculation of the electronic structure to further elucidate the catalytic mechanism.

Reviewer #2

(Remarks to the Author)

The paper from Sumner and coworkers addresses the ECH process of three model compounds: benzoic acid, phenol and guaiacol. They explore the experimental space by changing ECH variables such as current density, substrate concentration and temperature, and check how these variables influence conversion and faradaic efficiency of the process.

First thing that catches the eye is that the paper is very poorly written.

It needs to be proof-read again to correct all typos and missing punctuation, and to check the references.

From a scientific point of view the paper is ok, if a bit unexciting. The catalytic system is already known and the ECH of benzoic acid already reported. The point of the paper is to study the ECH of the complex mixture, but it stops at acknowledging the existence of complex interactions between the mixture components. I appreciate the difficulty of this type of study, but as is its scope is very limited.

Anyhow, the results are well presented and clear. Some details in the experimental part are missing (for example, I did not find anywhere if the catalyst is commercial or not and how the electrode is prepared) and do not allow reproducibility, but this is easily solvable. All in all the study is well done, maybe just very preliminary?

Reviewer #3

(Remarks to the Author)

The manuscript systematically investigated the electrochemical hydrogenation of benzoic acid, phenol, and guaiacol under different conditions. The use of GC-MS to detect intermediates provides valuable insights into the reaction pathways. With a few additional data, the work would be ready for publication.

1. In this manuscript, authors used GC-MS to investigate the reaction pathways. There is no CV data in the manuscript, which can provide reduction/oxidation peaks for understanding or verifying the proposed reaction pathways (Figure 5).
2. For the LSV curves in this manuscript, the author scanned from -0.5 V to ~-0.12 V vs. Ag/AgCl and then compared the voltage difference at -10 mA/cm². Adding a reverse LSV scan from ~-0.12 to -0.5 V vs. Ag/AgCl would help us to understand

the reaction dynamics, such as the reactions overpotential and Tafel slope.

3. The EIS measurements would help us to understand the configuration's charge transfer resistance and overall electrochemical performance. Including additional EIS data can help us to compare the reaction kinetics over different compounds.

Version 1:

Reviewer comments:

Reviewer #2

(Remarks to the Author)

I found the paper much improved after the review, as all the points raised by the reviewers were addressed by the authors. After their clarification on the importance of the work I understood better the scope and motivation of the work and I now appreciate the paper much more.

It is now suitable for publication as is.

Reviewer #3

(Remarks to the Author)

The authors have comprehensively responded to the reviewers' comments and suggestions. I recommend it for acceptance.

Response to the Reviewer's Comments

Reviewer's comment:	Authors' response
Reviewer 1	
Efficient and selective electrochemical hydrogenation (ECH) of organic compounds is critical to reducing global dependence on fossil fuels. In this work, the authors used a PtRu/ACC catalyst for the mild hydrogenation of bio-oil model compounds. BA+P showed the highest conversion (64.19%) and faraday efficiency (74%). In addition, the selectivity of BA to cyclohexanecarboxylic acid (CCA) was consistently maintained at 100% regardless of the experimental parameters. However, the quality of the manuscript remains to be improved. Some major comments have been listed for the authors' reference to further improve the quality of this work.	We appreciate this comment from the reviewer. Below please see our responses to the reviewer's specific comments:
1. The authors tested a variety of mixtures for conversion efficiency and faraday efficiency. It is recommended that the authors perform multiple experiments to take an average and establish error bars to further improve the accuracy of the data.	All experiments were performed in triplicates and the range of data were indicated using error bars. This have now been clarified in the text.
2. In this paper, there are few characterizations for the PtRu/ACC catalyst. (1) What is the atomic ratio of Pt and Ru for PtRu/ACC. This is crucial for the later modeling. (2) The XRD test is required to explore whether the material is in a pure phase or not. (3) The valence state of PtRu/ACC was characterized to further explain the catalyst mechanism. In addition to the above characterization, the authors still lack TEM, Raman, etc.	We appreciate these suggestions from the reviewer. All suggestions have been addressed in the revised manuscript, as detailed below: (1) The atomic ratio is 1:1 Pt:Ru, this was added to "DFT calculation" in both the results and methodology.(2) The XRD spectra was added to the manuscript (Figure 10g).(3) Raman analysis was added to the manuscript (Figure 10h).
3. The authors chose the (111) plane of PtRu/ACC for calculation. Please explain why this plane was chosen instead of other crystal planes such as (110).	This plane was chosen to maintain cohesion with the reference utilised. The authors agree that the plane orientation will have an impact on the bonding energy. Looking at this impact will make an interesting future study but is beyond the scope of the current paper.

Response to the Reviewer's Comments

4. The authors only calculate the surface energy, and lack the calculation for the electronic structure of the material. It is suggested that the authors supplement the calculation of the electronic structure to further elucidate the catalytic mechanism.	The electronic structure was calculated, and the charge density difference was added to the manuscript (Figure 8).
Reviewer 2	
The paper from Sumner and coworkers addresses the ECH process of three model compounds: benzoic acid, phenol and guaiacol. They explore the experimental space by changing ECH variables such as current density, substrate concentration and temperature, and check how these variables influence conversion and faradaic efficiency of the process. First thing that catches the eye is that the paper is very poorly written. It needs to be proof-read again to correct all typos and missing punctuation, and to check the references. From a scientific point of view the paper is ok, if a bit unexciting. The catalytic system is already known and the ECH of benzoic acid already reported. The point of the paper is to study the ECH of the complex mixture, but it stops at acknowledging the existence of complex interactions between the mixture components. I appreciate the difficulty of this type of study, but as its scope is very limited.	We thank the reviewer for his comment. The paper went through another round of proofreading to ensure it meets the standard. The novelty of this study lies in its investigation of the electrochemical hydrogenation (ECH) of mixed aromatic compounds using a bimetallic PtRu catalyst on activated carbon cloth, which effectively models the composition of bio-oil. By exploring the co-hydrogenation of benzoic acid, phenol, and guaiacol, the study reveals distinct interactions among these compounds, including both synergistic and antagonistic effects, impacting conversion rates and selectivity, including the formation of different products. Thus, this work serves as a link between the model compound and whole bio-oil studies, ultimately advancing our understanding of bio-oil upgrading under mild, sustainable conditions.
1. Anyhow, the results are well presented and clear. Some details in the experimental part are missing (for example, I did not find anywhere if the catalyst is commercial or not and how the electrode is prepared) and do not allow reproducibility, but this is easily solvable. All in all the study is well done, maybe just very preliminary?	The suppliers for the catalyst and other materials used in this study are summarised in the Materials. We appreciate the reviewer's comments regarding the quality assurance of this work. However, we would like to emphasise the significance and rigor of this study in response to the comment that this study being 'preliminary'. In addition to the novelty highlighted above, our research adopts a comprehensive experimental approach, including detailed electrochemical hydrogenation (ECH) tests on oxygenated aromatic compounds (OACs) and their mixtures. The manuscript provides a thorough justification for the selection of these compounds. Furthermore, we performed an extensive set of material

Response to the Reviewer's Comments

	characterisation techniques, and a factorial experimental design was employed to elucidate the interactions among the OACs. This experimental work is further supported and validated by density functional theory (DFT) calculations, which deepen our understanding of the reaction pathways and mechanisms of these aromatic compounds. We believe that this paper presents a rigorously executed experimental programme that makes a significant contribution to the field of ECH. Therefore, we respectfully disagree with the characterisation of this study as 'preliminary'.
Reviewer 3	
The manuscript systematically investigated the electrochemical hydrogenation of benzoic acid, phenol, and guaiacol under different conditions. The use of GC-MS to detect intermediates provides valuable insights into the reaction pathways. With a few additional data, the work would be ready for publication.	We appreciate this comment from the reviewer. Below please see our responses to the reviewer's specific comments:
1. In this manuscript, authors used GC-MS to investigate the reaction pathways. There is no CV data in the manuscript, which can provide reduction/oxidation peaks for understanding or verifying the proposed reaction pathways (Figure 5).	The CV data was added into the manuscript (Figure 11a).
2. For the LSV curves in this manuscript, the author scanned from -0.5 V to \sim-0.12 V vs. Ag/AgCl and then compared the voltage difference at -10 mA/cm². Adding a reverse LSV scan from \sim-0.12 to -0.5 V vs. Ag/AgCl would help us to understand the reaction dynamics, such as the reactions overpotential and Tafel slope	The LSV scan was carried out from -1.0V to 1.0V, the figure is a zoomed in image to focus on the current density of interest. The full figure was added into the manuscript to increase clarity. As the couple LSV + reverse LSV showed no difference from the CV curves, the reverse LSV data was omitted from the figure.
3. The EIS measurements would help us to understand the configuration's charge transfer resistance and overall electrochemical performance. Including additional EIS data can help us to compare the reaction kinetics over different compounds.	EIS data (Nyquist plot) was added to the manuscript (Figure 11d-f). Where: (d) EIS and zoomed in version (e) highlighting the baseline curve. (f) Equivalent circuit for EIS fitting